# Exploration via Hindsight Goal Generation

**Zhizhou Ren**[†]**, Kefan Dong**[†]
Institute for Interdisciplinary Information Sciences, Tsinghua University
Department of Computer Science, University of Illinois at Urbana-Champaign
{rzz16, dkf16}@mails.tsinghua.edu.cn

**Yuan Zhou**
Department of Industrial and Enterprise Systems Engineering
University of Illinois at Urbana-Champaign
yuanz@illinois.edu

**Qiang Liu**
Department of Computer Science
University of Texas at Austin
lqiang@cs.utexas.edu

**Jian Peng**
Department of Computer Science
University of Illinois at Urbana-Champaign
jianpeng@illinois.edu

## Abstract

Goal-oriented reinforcement learning has recently been a practical framework for robotic manipulation tasks, in which an agent is required to reach a certain goal defined by a function on the state space. However, the sparsity of such reward definition makes traditional reinforcement learning algorithms very inefficient. Hindsight Experience Replay (HER), a recent advance, has greatly improved sample efficiency and practical applicability for such problems. It exploits previous replays by constructing imaginary goals in a simple heuristic way, acting like an implicit curriculum to alleviate the challenge of sparse reward signal. In this paper, we introduce Hindsight Goal Generation (HGG), a novel algorithmic framework that generates valuable hindsight goals which are easy for an agent to achieve in the short term and are also potential for guiding the agent to reach the actual goal in the long term. We have extensively evaluated our goal generation algorithm on a number of robotic manipulation tasks and demonstrated substantially improvement over the original HER in terms of sample efficiency.

## 1   Introduction

Recent advances in deep reinforcement learning (RL), including policy gradient methods (Schulman et al., 2015, 2017) and Q-learning (Mnih et al., 2015), have demonstrated a large number of successful applications in solving hard sequential decision problems, including robotics (Levine et al., 2016), games (Silver et al., 2016; Mnih et al., 2015), and recommendation systems (Karatzoglou et al., 2013), among others. To train a well-behaved policy, deep reinforcement learning algorithms use neural networks as functional approximators to learn a state-action value function or a policy distribution to optimize a long-term expected return. The convergence of the training process, particularly in Q-learning, is heavily dependent on the temporal pattern of the reward function (Szepesvári, 1998). For example, if only a non-zero reward/return is provided at the end of an rollout of a trajectory with length $L$, while no rewards are observed before the $L$-th time step, the Bellman updates of the Q-function would become very inefficient, requiring at least $L$ steps to propagate the final return to the

---

[†]Work done while Zhizhou and Kefan were visiting students at UIUC.

Q-function of all earlier state-action pairs. Such sparse or episodic reward signals are ubiquitous in many real-world problems, including complex games and robotic manipulation tasks (Andrychowicz et al., 2017). Therefore, despite its notable success, the application of RL is still quite limited to real-world problems, where the reward functions can be sparse and very hard to engineer (Ng et al., 1999). In practice, human experts need to design reward functions which would reflect the task needed to be solved and also be carefully shaped in a dense way for the optimization in RL algorithms to ensure good performance. However, the design of such dense reward functions is non-trivial in most real-world problems with sparse rewards. For example, in goal-oriented robotics tasks, an agent is required to reach some state satisfying predefined conditions or within a state set of interest. Many previous efforts have shown that the sparse indicator rewards, instead of the engineered dense rewards, often provide better practical performance when trained with deep Q-learning and policy optimization algorithms (Andrychowicz et al., 2017). In this paper, we will focus on improving training and exploration for goal-oriented RL problems.

A notable advance is called *Hindsight Experience Replay (HER)* (Andrychowicz et al., 2017), which greatly improves the practical success of off-policy deep Q-learning for goal-oriented RL problems, including several difficult robotic manipulation tasks. The key idea of HER is to revisit previous states in the experience replay and construct a number of achieved hindsight goals based on these visited intermediate states. Then the hindsight goals and the related trajectories are used to train an universal value function parameterized by a goal input by algorithms such as deep deterministic policy gradient (DDPG, Lillicrap et al. (2016)). A good way to think of the success of HER is to view HER as an implicit curriculum which first learns with the intermediate goals that are easy to achieve using current value function and then later with the more difficult goals that are closer to the final goal. A notable difference between HER and curriculum learning is that HER does not require an explicit distribution of the initial environment states, which appears to be more applicable to many real problems.

In this paper, we study the problem of automatically generating valuable hindsight goals which are more effective for exploration. Different from the random curriculum heuristics used in the original HER, where a goal is drawn as an achieved state in a random trajectory, we propose a new approach that finds intermediate goals that are easy to achieve in the short term and also would likely lead to reach the final goal in the long term. To do so, we first approximate the value function of the actual goal distribution by a lower bound that decomposes into two terms, a value function based on a hindsight goal distribution and the Wasserstein distance between the two distributions. Then, we introduce an efficient discrete Wasserstein Barycenter solver to generate a set of hindsight goals that optimizes the lower bound. Finally, such goals are used for exploration.

In the experiments, we evaluate our Hindsight Goal Generation approach on a broad set of robotic manipulation tasks. By incorporating the hindsight goals, a significant improvement on sample efficiency is demonstrated over DDPG+HER. Ablation studies show that our exploration strategy is robust across a wide set of hyper-parameters.

## 2  Background

**Reinforcement Learning** The goal of reinforcement learning agent is to interact with a given environment and maximize its expected cumulative reward. The environment is usually modeled by a Markov Decision Process (MDP), given by tuples $\langle \mathcal{S}, \mathcal{A}, P, R, \gamma \rangle$, where $\mathcal{S}, \mathcal{A}$ represent the set of states and actions respectively. $P : \mathcal{S} \times \mathcal{A} \to \mathcal{S}$ is the transition function and $R : \mathcal{S} \times \mathcal{A} \to [0, 1]$ is the reward function. $\gamma$ is the discount factor. The agent trys to find a policy $\pi : \mathcal{S} \to \mathcal{A}$ that maximizes its expected curriculum reward $V^\pi(s_0)$, where $s_0 = s$ is usually given or drawn from a distribution $\mu_0$ of initial state. The value function $V^\pi(s)$ is defined as

$$V^\pi(s) = \mathbb{E}_{s_0=s, a_t \sim \pi(\cdot|s_t), s_{t+1} \sim P(\cdot|s_t, a_t)} \left[ \sum_{t=0}^{\infty} \gamma^t R(s_t, a_t) \right].$$

**Goal-oriented MDP** In this paper, we consider a specific class of MDP called goal-oriented MDP. We use $\mathcal{G}$ to denote the set of goals. Different from traditional MDP, the reward function $R$ is a goal-conditioned sparse and binary signal indicating whether the goal is achieved:

$$R_g(s_t, a_t, s_{t+1}) := \begin{cases} 0, & \|\phi(s_{t+1}) - g\|_2 \leq \delta_g \\ -1, & \text{otherwise.} \end{cases} \tag{1}$$

$\phi : \mathcal{S} \to \mathcal{G}$ is a known and tractable mapping that defines goal representation. $\delta_g$ is a given threshold indicating whether the goal is considered to be reached (see Plappert et al. (2018)).

**Universal value function** The idea of universal value function is to use a single functional approximator, such as neural networks, to represent a large number of value functions. For the goal-oriented MDPs, the goal-based value function of a policy $\pi$ for any given goal $g$ is defined as $V^\pi(s, g)$, for all state $s \in \mathcal{S}$. That is

$$V^\pi(s, g) := \mathbb{E}_{s_0 = s, a_t \sim \pi(\cdot|s_t, g), s_{t+1} \sim P(\cdot|s_t, a_t)} \left[ \sum_{t=0}^{\infty} \gamma^t R_g(s_t, a_t, s_{t+1}) \right]. \tag{2}$$

Let $\mathcal{T}^* : \mathcal{S} \times \mathcal{G} \to [0, 1]$ be the joint distribution over starting state $s_0 \in \mathcal{S}$ and goal $g \in \mathcal{G}$.. That is, at the start of every episode, a state-goal pair $(s_0, g)$ will be drawn from the task distribution $\mathcal{T}^*$. The agent tries to find a policy $\pi : \mathcal{S} \times \mathcal{G} \to \mathcal{A}$ that maximizes the expectation of discounted cumulative reward

$$V^\pi(\mathcal{T}^*) := \mathbb{E}_{(s_0, g) \sim \mathcal{T}^*} [V^\pi(s_0, g)] \tag{3}$$

Goal-oriented MDP characterizes several reinforcement benchmark tasks, such as the robotics tasks in the OpenAI gym environment (Plappert et al., 2018). For example, in the FetchPush (see Figure 1) task, the agent needs to learn pushing a box to a designated point. In this task, the state of the system $s$ contains the status for both the robot and the box. The goal $g$, on the other hand, only indicates the designated position of the box. Thus, the mapping $\phi$ is defined as a mapping from a system state $s$ to the position of the box in $s$.

**Access to Simulator** One of the common assumption made by previous work is an universal simulator that allows the environment to be reset to any given state (Florensa et al., 2017; Ecoffet et al., 2019). This kind of simulator is excessively powerful, and hard to build when acting in the real world. On the contrary, our method does not require an universal simulator, and thus is more realizable.

## 3 Related Work

**Multi-Goal RL** The role of goal-conditioned policy has been investigated widely in deep reinforcement learning scenarios (Pong et al., 2019). A few examples include grasping skills in imitation learning (Pathak et al., 2018; Srinivas et al., 2018), disentangling task knowledge from environment (Mao et al., 2018a; Ghosh et al., 2019), and constituting lower-level controller in hierarchical RL (Oh et al., 2017; Nachum et al., 2018; Huang et al., 2019; Eysenbach et al., 2019). By learning a universal value function which parameterizes the goal using a function approximator (Schaul et al., 2015), an agent is able to learn multiple tasks simultaneously (Kaelbling, 1993; Veeriah et al., 2018) and identify important decision states (Goyal et al., 2019b). It is shown that multi-task learning with goal-conditioned policy improves the generalizability to unseen goals (e.g., Schaul et al. (2015)).

**Hindsight Experience Replay** Hindsight Experience Replay (Andrychowicz et al., 2017) is an effective experience replay strategy which generates reward signal from failure trajectories. The idea of hindsight experience replay can be extended to various goal-conditioned problems, such as hierarchical RL (Levy et al., 2019), dynamic goal pursuit (Fang et al., 2019a), goal-conditioned imitation (Ding et al., 2019; Sun et al., 2019) and visual robotics applications (Nair et al., 2018; Sahni et al., 2019). It is also shown that hindsight experience replay can be combined with on-policy reinforcement learning algorithms by importance sampling (Rauber et al., 2019).

**Curriculum Learning in RL** Curriculum learning in RL usually suggests using a sequence of auxiliary tasks to guide policy optimization, which is also related to multi-task learning, lifelong learning, and transfer learning. The research interest in automatic curriculum design has seen rapid growth recently, where approaches have been proposed to schedule a given set of auxiliary tasks (Riedmiller et al., 2018; Colas et al., 2019), and to provide intrinsic motivation (Forestier et al., 2017; Péré et al., 2018; Sukhbaatar et al., 2018; Colas et al., 2018). Generating goals which leads to high-value states could substantially improve the sample efficiency of RL agent (Goyal et al., 2019a). Guided exploration through curriculum generation is also an active research topic, where either the initial state (Florensa et al., 2017) or the goal position (Baranes and Oudeyer, 2013; Florensa et al., 2018) is considered as a manipulable factor to generate the intermediate tasks. However, most

curriculum learning methods are domain-specific, and it is still open to build a generalized framework for curriculum learning.

# 4   Automatic Hindsight Goal Generation

As discussed in the previous section, HER provides an effective solution to resolve the sparse reward challenge in object manipulation tasks, in which achieved state in some past trajectories will be replayed as imaginary goals. In the other words, HER modifies the task distribution in replay buffer to generate a set of auxiliary nearby goals which can used for further exploration and improve the performance of an off-policy RL agent which is expected to reach a very distant goal. However, the distribution of hindsight goals where the policy is trained on might differ significantly from the original task or goal distribution. Take Figure 1 as an example, the desired goal distribution is lying on the red segment, which is far away from the initial position. In this situation, those hindsight

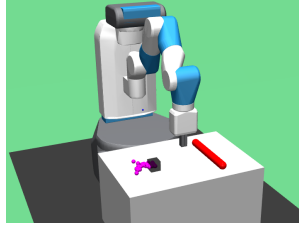

Figure 1: Visualization of hindsight goals (pink particles).

goals may not be effective enough to promote policy optimization in original task. The goal of our work is to develop a new approach to generate valuable hindsight goals that will improve the performance on the original task.

In the rest of this section, we will present a new algorithmic framework as well as our implementation for automatic hindsight goal generation for better exploration.

## 4.1   Algorithmic Framework

Following Florensa et al. (2018), our approach relies on the following generalizability assumption.

**Assumption 1.** *A value function of a policy $\pi$ for a specific goal $g$ has some generalizability to another goal $g'$ close to $g$.*

One possible mathematical characterization for Assumption 1 is via the Lipschitz continuity. Similar assumptions have been widely applied in many scenarios (Asadi et al., 2018; Luo et al., 2019):

$$|V^\pi(s,g) - V^\pi(s',g')| \leq L \cdot d((s,g),(s',g')), \tag{4}$$

where $d((s,g),(s',g'))$ is a metric defined by

$$d((s,g),(s',g')) = c\|\phi(s) - \phi(s')\|_2 + \|g - g'\|_2. \tag{5}$$

for some hyperparameter $c > 0$ that provides a trade-off between the distances between initial states and the distance between final goals. $\phi(\cdot)$ is a state abstraction to map from the state space to the goal space. When experimenting with the tasks in the OpenAI Gym environment (Plappert et al., 2018), we simply adopt the state-goal mappings as defined in (1). Although the Lipschitz continuity may not hold for every $s, s' \in \mathcal{S}, g, g' \in \mathcal{G}$, we only require continuity over some specific region. It is reasonable to claim that bound Eq. (4) holds for most of the $(s, g), (s', g')$ when $d((s,g),(s',g'))$ is not too large.

Partly due to the reward sparsity of the distant goals, optimizing the expected cumulative reward (see Eq. (3)) from scratch is very difficult. Instead, we propose to optimize a relaxed lower bound which introduces intermediate goals that may be easier to optimize. Here we provide Theorem 1 that establishes the such a lower bound.

**Theorem 1.** *Assuming that the generalizability condition (Eq. (4)) holds for two distributions $(s, g) \sim \mathcal{T}$ and $(s', g') \sim \mathcal{T}'$, we have*

$$V^\pi(\mathcal{T}') \geq V^\pi(\mathcal{T}) - L \cdot D(\mathcal{T}, \mathcal{T}'). \tag{6}$$

*where $D(\cdot, \cdot)$ is the Wasserstein distance based on $d(\cdot, \cdot)$*

$$D(\mathcal{T}^{(1)}, \mathcal{T}^{(2)}) = \inf_{\mu \in \Gamma(\mathcal{T}^{(1)}, \mathcal{T}^{(2)})} \left( \mathbb{E}_\mu[d((s_0^{(1)}, g^{(1)}), (s_0^{(2)}, g^{(2)}))] \right)$$

*where $\Gamma(\mathcal{T}^{(1)}, \mathcal{T}^{(2)})$ denotes the collection of all joint distribution $\mu(s_0^{(1)}, g^{(1)}, s_0^{(2)}, g^{(2)})$ whose marginal probabilities are $\mathcal{T}^{(1)}, \mathcal{T}^{(2)}$, respectively.*

The proof of Theorem 1 is deferred to Appendix A.

It follows from Theorem 1 that optimizing cumulative rewards Eq. (3) can be relaxed into the following surrogate problem

$$\max_{\mathcal{T},\pi} \quad V^\pi(\mathcal{T}) - L \cdot D(\mathcal{T}, \mathcal{T}^*). \tag{7}$$

Note that this new objective function is very intuitive. Instead of optimizing with the difficult goal/task distribution $\mathcal{T}^*$, we hope to find a collection of surrogate goals $\mathcal{T}$, which are both easy to optimize and are also close or converging towards $\mathcal{T}^*$. However the joint optimization of $\pi$ and $\mathcal{T}$ is non-trivial. This is because a) $\mathcal{T}$ is a high-dimensional distribution over tasks, b) policy $\pi$ is optimized with respect to a shifting task distribution $\mathcal{T}$, c) the estimation of value function $V^\pi(\mathcal{T})$ may not be quite accurate during training.

Inspired by Andrychowicz et al. (2017), we adopt the idea of using hindsight goals here. We first enforce $\mathcal{T}$ to be a finite set of $K$ particles which can only be from those already achieved states/goals from the replay buffer $B$. In another word, the support of the set $\mathcal{T}$ should lie inside $B$. In the meanwhile, we notice that a direct implementation of problem Eq. (7) may lead to degeneration of hindsight goal selection of the training process, i.e., the goals may be all drawn from a single trajectory, thus not being able to provide sufficient exploration. Therefore, we introduce an extra diversity constraint, i.e, for every trajectory $\tau \in B$, at most $\mu$ states can be selected in $\mathcal{T}$. In practice, we find that simply setting it to 1 would result in reasonable performance. It is shown in Section 5.3 that this diversity constraint indeed improves the robustness of our algorithm.

Finally, the optimization problem we aim to solve is,

$$\max_{\pi,\mathcal{T}:|\mathcal{T}|=K} \quad V^\pi(\mathcal{T}) - L \cdot D(\mathcal{T}, \mathcal{T}^*)$$

$$\text{s.t.} \quad \sum_{s_0,s_t \in \tau} \mathbb{1}[(s_0, \phi(s_t)) \in \mathcal{T}] \leq 1, \quad \forall \tau \in B$$

$$\sum_{\tau \in B} \sum_{s_0,s_t \in \tau} \mathbb{1}[(s_0, \phi(s_t)) \in \mathcal{T}] = K.$$

To solve the above optimization, we adapt a two-stage iterative algorithm. First, we apply a policy optimization algorithm, for example DDPG, to maximize the value function conditioned on the task set $\mathcal{T}$. Then we fix $\pi$ and optimize the the hindsight set $\mathcal{T}$ subject to the diversity constraint, which is a variant of the well-known Wasserstein Barycenter problem with a bias term (the value function) for each particle. Then we iterate the above process until the policy achieves a desirable performance or we reach a computation budget. It is not hard to see that the first optimization of value function is straightforward. In our work, we simply use the DDPG+HER framework for it. The second optimization of hindsight goals is non-trivial. In the following, we describe an efficient approximation algorithm for it.

## 4.2 Solving Wasserstein Barycenter Problem via Bipartite Matching

Since we assume that $\mathcal{T}$ is hindsight and with $K$ particles, we can approximately solve the above Wasserstein Barycenter problem in the combinatorial setting as a bipartite matching problem. Instead of dealing with $\mathcal{T}^*$, we draw $K$ samples from $\mathcal{T}^*$ to empirically approximate it by a set of $K$ particles $\widehat{\mathcal{T}}^*$. In this way, the hindsight task set $\mathcal{T}$ can be solved in the following way. For every task instance $(\hat{s}_0^i, \hat{g}^i) \in \widehat{\mathcal{T}}^*$, we find a state trajectory $\tau^i = \{s_t^i\} \in B$ that together minimizes the sum

$$\sum_{(\hat{s}_0^i, \hat{g}^i) \in \widehat{\mathcal{T}}^*} w((\hat{s}_0^i, \hat{g}^i), \tau^i) \tag{8}$$

where we define

$$w((\hat{s}_0^i, \hat{g}^i), \tau^i) := c\|\phi(\hat{s}_0^i) - \phi(s_0^i)\|_2 + \min_t \left( \|\hat{g}^i - \phi(s_t^i)\|_2 - \frac{1}{L} V^\pi(s_0^i, \phi(s_t^i)) \right). \tag{9}$$

Finally we select each corresponding achieved state $s_t \in \tau$ to construct hindsight goal $(\hat{s}_0, \phi(s_t)) \in \mathcal{T}$. It is not hard to see that the above combinatorial optimization exactly identifies optimal solution

$\mathcal{T}$ in the above-mentioned Wasserstein Barycenter problem. In practice, the Lipschitz constant $L$ is unknown and therefore treated as a hyper-parameter.

The optimal solution of the combinatorial problem in Eq. (8) can be solved efficiently by the well-known maximum weight bipartite matching (Munkres, 1957; Duan and Su, 2012). The bipartite graph $G(\{V_x, V_y\}, E)$ is constructed as follows. Vertices are split into two partitions $V_x, V_y$. Every vertex in $V_x$ represents a task instance $(\hat{s}_0, \hat{g}) \in \hat{\mathcal{T}}^*$, and vertex in $V_y$ represents a trajectory $\tau \in B$. The weight of edge connecting $(\hat{s}_0, \hat{g})$ and $\tau$ is $-w((\hat{s}_0, \hat{g}), \tau)$ as defined in Eq. (9). In this paper, we apply the Minimum Cost Maximum Flow algorithm to solve this bipartite matching problem (for example, see Ahuja et al. (1993)).

---

**Algorithm 1** Exploration via Hindsight Goal Generation (HGG)

---

1: Initialize $\pi$          ▷ initialize neural networks
2: $B \leftarrow \emptyset$
3: **for** iteration $= 1, 2, \ldots, N$ **do**
4:      Sample $\{(\hat{s}_0^i, \hat{g}^i)\}_{i=1}^K \sim \mathcal{T}^*$       ▷ sample from target distribution
5:      Find $K$ distinct trajectories $\{\tau^i\}_{i=1}^K$ that minimize       ▷ weighted bipartite matching

$$\sum_{i=1}^K w((\hat{s}_0^i, \hat{g}^i), \tau^i) = \sum_{i=1}^K \left( c\|\phi(\hat{s}_0^i) - \phi(s_0^i)\|_2 + \min_t \left( \|\hat{g}^i - \phi(s_t^i)\|_2 - \frac{1}{L} V^\pi(s_0^i, \phi(s_t^i)) \right) \right)$$

6:      Construct intermediate task distribution $\{(\hat{s}_0^i, g^i)\}_{i=1}^M$ where

$$g^i = \phi \left( \underset{s_t^i \in \tau_i}{\arg\min} \left( \|\hat{g}^i - \phi(s_t^i)\|_2 - \frac{1}{L} V^\pi(s_0^i, \phi(s_t^i)) \right) \right)$$

7:      **for** $i = 1, 2, \ldots, K$ **do**
8:          $(s_0, g) \leftarrow (\hat{s}_0^i, g^i)$       ▷ **critical step: hindsight goal-oriented exploration**
9:          **for** $t = 0, 1, \ldots, H - 1$ **do**
10:            $a_t \leftarrow \pi(\cdot|s_t, g) + \text{noise}$       ▷ together with $\epsilon$-greedy or Gaussian exploration
11:            $s_{t+1} \sim P(\cdot|s_t, a_t)$
12:            $r_t \leftarrow R_g(s_t, a_t, s_{t+1})$
13:          $\tau \leftarrow \{s_0, a_0, r_0, s_1, \ldots\}$
14:          $B \leftarrow B \cup \{\tau\}$
15:      **for** $i = 1 \ldots M$ **do**
16:          Sample a minibatch $b$ from replay buffer using HER
17:          Perform one step on value and policy update on minibatch $b$ using DDPG

---

**Overall Algorithm** The overall description of our algorithm is shown in Algorithm 1. Note that our exploration strategy the only modification is in Step 8, in which we generate hindsight goals to guide the agent to collect more valuable trajectories. So it is complementary to other improvements in DDPG/HER around Step 16, such as the prioritized experience replay strategy (Schaul et al., 2016; Zhao and Tresp, 2018; Zhao et al., 2019) and other variants of hindsight experience replay (Fang et al., 2019b; Bai et al., 2019).

## 5 Experiments

Our experiment environments are based on the standard robotic manipulation environments in the OpenAI Gym (Brockman et al., 2016)[3]. In addition to the standard settings, to better visualize the improvement of the sample efficiency, we vary the target task distributions in the following ways:

- Fetch environments: Initial object position and goal are generated uniformly at random from two distant segments.
- Hand-manipulation environments : These tasks require the agent to rotate the object into a given pose, and only the rotations around $z$-axis are considered here. We restrict the initial

axis-angle in a small interval, and the target pose will be generated in its symmetry. That is, the object needs to be rotated in about $\pi$ degree.

- Reach environment: FetchReach and HandReach do not support randomization of the initial state, so we restrict their target distribution to be a subset of the original goal space.

Regarding baseline comparison, we consider the original DDPG+HER algorithm. We also investigate the integration of the experience replay prioritization strategies, such as the Energy-Based Prioritization (EBP) proposed by Zhao and Tresp (2018), which draws the prior knowledge of physics system to exploit valuable trajectories. More details of experiment settings are included in the Appendix B.

## 5.1 HGG Generates Better Hindsight Goals for Exploration

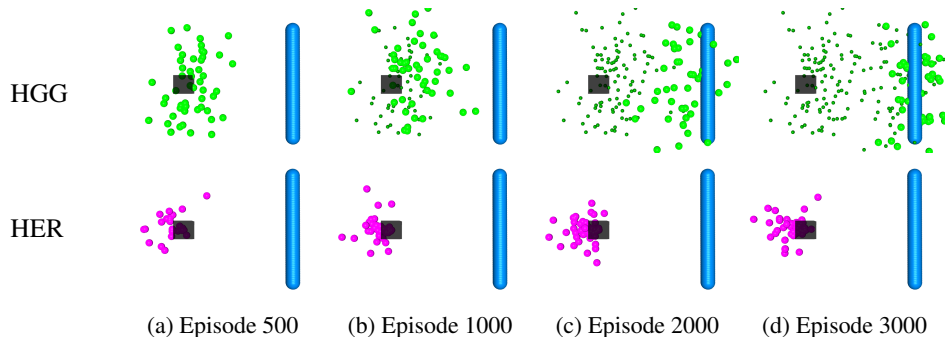

(a) Episode 500　　(b) Episode 1000　　(c) Episode 2000　　(d) Episode 3000

Figure 2: Visualization of goal distribution generated by HGG and HER on FetchPush. The initial object position is shown as a black box. The blue segment indicates target goal distribution. The above row presents the distribution of the hindsight goals generated by our HGG method, where bright green particles is a batch of recently generated goals, and dark green particles present the goals generated in the previous iterations. The bottom row presents the distribution of replay goals generated by HER.

We first check whether HGG is able to generate meaningful hindsight goals for exploration. We compare HGG and HER in the FetchPush environment. It is shown in Figure 2 that HGG algorithm generates goals that gradually move towards the target region. Since those goals are hindsight, they are considered to be achieved during training. In comparison, the replay distribution of a DDPG+HER agent has been stuck around the initial position for many iterations, indicating that those goals may not be able to efficiently guide exploration.

**Performance on benchmark robotics tasks**

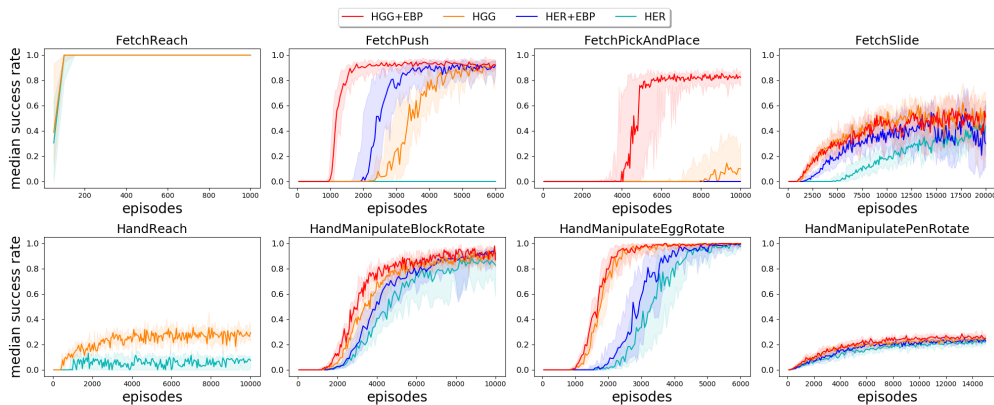

Figure 3: Learning curves for variant a number of goal-oriented robotic manipulation tasks. All curves presented in this figure are trained with default hyper-parameters included in Appendix C.1. Note that since FetchReach and HandReach do not contain object instances for EBP, so we do not include the +EBP versions for them.

Then we check whether the exploration provided by the goals generated by HGG can result in better policy training performance. As shown in Figure 3, we compare the vanilla HER, HER with Energy-Based Prioritization (HER+EBP), HGG, HGG+EBP. It is worth noting that since EBP is designed for the Bellman equation updates, it is complementary to our HGG-based exploration approach. Among the eight environments, HGG substantially outperforms HER on four and has comparable performance on the other four, which are either too simple or too difficult. When combined with EBP, HGG+EBP achieves the best performance on six environments that are eligible.

**Performance on tasks with obstacle** In a more difficult task, crafted metric may be more suitable than $\ell_2$-distance used in Eq. (5). As shown in Figure 4, we created an environment based on FetchPush with a rigid obstacle. The object and the goal are uniformly generated in the green and the red segments respectively. The brown block is a static wall which cannot be moved. In addition to $\ell_2$, we also construct a distance metric based on the graph distance of a mesh

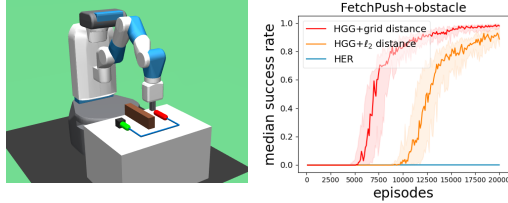

Figure 4: Visualization of FetchPush with obstacle.

grid on the plane, the blue line is a successful trajectory in such hand-craft distance measure. A more detailed description is deferred to Appendix B.3. Intuitively speaking, this crafted distance should be better than $\ell_2$ due to the existence of the obstacle. Experimental results suggest that such a crafted distance metric provides better guidance for goal generation and training, and significantly improves sample efficiency over $\ell_2$ distance. It would be a future direction to investigate ways to obtain or learn a good metric.

## 5.2 Comparison with Explicit Curriculum Learning

Since our method can be seen as an explicit curriculum learning for exploration, where we generate hindsight goals as intermediate task distribution, we also compare our method with another recently proposed curriculum learning method for RL. Florensa et al. (2018) leverages Least-Squares GAN (Mao et al., 2018b) to mimic the set called Goals of Intermediate Difficult as exploration goal generator.

Specifically, in our task settings, we define a goal set $GOID(\pi) = \{g : \alpha \leq f(\pi, g) \leq 1 - \alpha\}$, where $f(\pi, g)$ represents the average success rate in a small region closed by goal $g$. To sample from $GOID$, we implement an oracle goal generator based on rejection sampling, which could uniformly sample goals from $GOID(\pi)$. Result in Figure 5 indicates

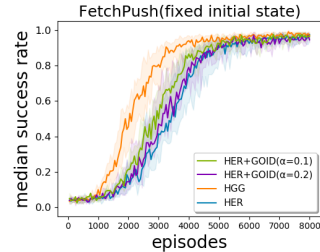

Figure 5: Comparison with curriculum learning. We compare HGG with the original HER, HER+GOID with two threshold values.

that our Hindsight Goal Generation substantially outperforms HER even with $GOID$ from the oracle generator. Note that this experiment is run on a environment with fixed initial state due to the limitation of Florensa et al. (2018). The choice of $\alpha$ is also suggested by Florensa et al. (2018).

## 5.3 Ablation Studies on Hyperparameter Selection

In this section, we set up a set of ablation tests on several hyper-parameters used in the Hindsight Goal Generation algorithm.

**Lipschitz $L$:** The selection of Lipschitz constant is task dependent, since it iss related with scale of value function and goal distance. For the robotics tasks tested in this paper, we find that it is easier to set $L$ by first divided it with the upper bound of the distance between any two final goals in a environment. We test a few choices of $L$ on several environments and find that it is very easy to find a range of $L$ that works well and shows robustness for all the environments tested in this section. We show the learning curves on FetchPush with different $L$. It appears that the performance of HGG is reasonable as long as $L$ is not too small. For all tasks we tested in the comparisons, we set $L = 5.0$.

**Distance weight $c$:** Parameter $c$ defines the trade-off between the initial state similarity and the goal similarity. Larger $c$ encourages our algorithm to choose hindsight goals that has closer initial state.

Results in Figure 6 indicates that the choice of $c$ is indeed robust. For all tasks we tested in the comparisons, we set $c = 3.0$.

**Number of hindsight goals $K$:** We find that for the simple tasks, the choice of $K$ is not critical. Even a greedy approach (corresponds to $K = 1$) can achieved competitive performance, e.g. on FetchPush in the third panel of Figure 6. For more difficult environment, such as FetchPickAndPlace, larger batch size can significantly reduce the variance of training results. For all tasks tested in the comparisons, we ploted the best results given by $K \in \{50, 100\}$.

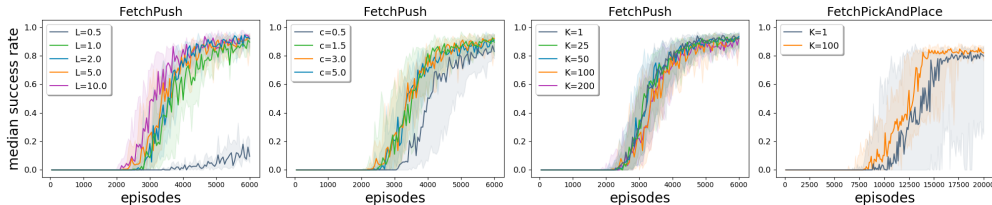

Figure 6: Ablation study of hyper-parameter selection. Several curves are omitted in the forth panel to provide a clear view of variance comparison. A full version is deferred to Appendix D.4.

# 6    Conclusion

We present a novel automatic hindsight goal generation algorithm, by which valuable hindsight imaginary tasks are generated to enable efficient exploration for goal-oriented off-policy reinforcement learning. We formulate this idea as a surrogate optimization to identify hindsight goals that are easy to achieve and also likely to lead to the actual goal. We introduce a combinatorial solver to generate such intermediate tasks. Extensive experiments demonstrated better goal-oriented exploration of our method over original HER and curriculum learning on a collection of robotic learning tasks. A future direction is to incorporate the controllable representation learning (Thomas et al., 2017) to provide task-specific distance metric (Ghosh et al., 2019; Srinivas et al., 2018), which may generalize our method to more complicated cases where the standard Wasserstein distance cannot be applied directly.

## Footnotes

[3]Our code is available at `https://github.com/Stilwell-Git/Hindsight-Goal-Generation`.

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
