[Supplementary Material · Exploration via Hindsight Goal Generation_SM.pdf]

# A Proof of Theorem 1

In this section we provide the proof of Theorem 1.

**Theorem 1.** *Assuming that the generalizability condition (Eq. (4)) holds for two distributions $(s,g) \sim \mathcal{T}$ and $(s',g') \sim \mathcal{T}'$, we have*

$$V^{\pi}(\mathcal{T}') \geq V^{\pi}(\mathcal{T}) - L \cdot D(\mathcal{T}, \mathcal{T}'). \qquad (6)$$

*where $D(\cdot, \cdot)$ is the Wasserstein distance based on $d(\cdot, \cdot)$*

$$D(\mathcal{T}^{(1)}, \mathcal{T}^{(2)}) = \inf_{\mu \in \Gamma(\mathcal{T}^{(1)}, \mathcal{T}^{(2)})} \left( \mathbb{E}_{\mu}[d((s_0^{(1)}, g^{(1)}), (s_0^{(2)}, g^{(2)}))] \right)$$

*where $\Gamma(\mathcal{T}^{(1)}, \mathcal{T}^{(2)})$ denotes the collection of all joint distribution $\mu(s_0^{(1)}, g^{(1)}, s_0^{(2)}, g^{(2)})$ whose marginal probabilities are $\mathcal{T}^{(1)}, \mathcal{T}^{(2)}$, respectively.*

*Proof.* By Eq. (4), for any quadruple $(s, g, s', g')$, we have

$$V^{\pi}(s', g') \geq V^{\pi}(s, g) - L \cdot d((s, g), (s', g')). \qquad (10)$$

For any $\mu \in \Gamma(\mathcal{T}, \mathcal{T}')$, we sample $(s, g, s', g') \sim \mu$ and take the expectation on both sides of Eq. (10), and get

$$V^{\pi}(\mathcal{T}') \geq V^{\pi}(\mathcal{T}) - L \cdot \mathbb{E}_{\mu}[d((s, g), (s', g'))]. \qquad (11)$$

Since Eq. (11) holds for any $\mu \in \Gamma(\mathcal{T}, \mathcal{T}')$, we have

$$V^{\pi}(\mathcal{T}') \geq V^{\pi}(\mathcal{T}) - L \cdot \inf_{\mu \in \Gamma(\mathcal{T}, \mathcal{T}')} (\mathbb{E}_{\mu}[d((s, g), (s', g'))]) = V^{\pi}(\mathcal{T}) - L \cdot D(\mathcal{T}, \mathcal{T}').$$

$\square$

# B Experiment Settings

## B.1 Modified Environments

Figure 7: Visualization of modified task distribution in Fetch environments. The object is uniformly generated on the green segment, and the goal is uniformly generated on the red segment.

Fetch Environments:

- FetchPush-v1: Let the origin $(0, 0, 0)$ denote the projection of gripper's initial coordinate on the table. The object is uniformly generated on the segment $(-0.15, -0.15, 0) - (0.15, -0.15, 0)$, and the goal is uniformly generated on the segment $(-0.15, 0.15, 0) - (0.15, 0.15, 0)$.

- FetchPickAndPlace-v1: Let the origin $(0, 0, 0)$ denote the projection of gripper's initial coordinate on the table. The object is uniformly generated on the segment $(-0.15, -0.15, 0) - (0.15, -0.15, 0)$, and the goal is uniformly generated on the segment $(-0.15, 0.15, 0.45) - (0.15, 0.15, 0.45)$.

- FetchSlide-v1: Let the origin $(0, 0, 0)$ denote the projection of gripper's initial coordinate on the table. The object is uniformly generated on the segment $(-0.05, -0.1, 0) - (-0.05, 0.1, 0)$, and the goal is uniformly generated on the segment $(0.55, -0.15, 0) - (0.55, 0.15, 0)$.

Hand Environments:

- HandManipulateBlockRotate-v0, HandManipulateEggRotate-v0: Let $s_0$ be the default initial state defined in original simulator (Plappert et al., 2018). The initial pose is generated by applying a rotation around $z$-axis, where the rotation degree will be uniformly sampled from $[-\pi/4, \pi/4]$. The goal is also rotated from $s_0$ around $z$-axis, where the degree is uniformly sampled from $[\pi - \pi/4, \pi + \pi/4]$.

- HandManipulatePenRotate-v0: We use the same setting as the original simulator.

Reach Environments:

- FetchReach-v1: Let the origin $(0, 0, 0)$ denote the coordinate of gripper's initial position. Goal is uniformly generated on the segment $(-0.15, 0.15, 0.15) - (0.15, 0.15, 0.15)$.

- HandReach-v0: Uniformly select one dimension of *meeting point* and add an offset of 0.005, where *meeting point* is defined in original simulator (Plappert et al., 2018)

Other attributes of the environment (such as horizon $H$, reward function $R_g$) are kept the same as default.

## B.2 Evaluation Details

- All curves presented in this paper are plotted from 10 runs with random task initializations and seeds.

- Shaded region indicates 60% population around median.

- All curves are plotted using the same hyper-parameters (except ablation section).

- Following Andrychowicz et al. (2017), an episode is considered successful if $\|\phi(s_H) - g\|_2 \leq \delta_g$ is achieved, where $\phi(s_H)$ is the object position at the end of the episode. $\delta_g$ is the same threshold using in reward function (1).

## B.3 Details of Experiment with obstacle

Using the same coordinate system as Appendix B.1. Let the origin $(0, 0, 0)$ denote the projection of gripper's initial coordinate on the table. The object is uniformly generated on the segment $(-0.15, -0.15, 0) - (-0.045, -0.15, 0)$, and the goal is uniformly generated on the segment $(-0.15, 0.15, 0) - (-0.045, 0.15, 0)$. The wall lies on $(-0.3, 0, 0) - (0, 0, 0)$.

The crafted distance used in Figure 4 is calculated by the following rules.

- The distance metric between two initial states is kept as before.

- The distance between the hindsight goal $g$ and the desired goal $g^*$ is evaluated as the summation of two parts. The first part is the $\ell_2$ distance between the goal $g$ and its closest point $g'$ on the blue polygonal line shown in Figure 4. The second part the distance between $g'$ and $g^*$ along the blue line.

- The above two terms are comined with the same ratio used in Eq. (5).

## B.4 Details of Experiment 5.2

- Since the environment is deterministic, the success rate $f(\pi, g)$ is defines as

$$f(\pi, g) = \int_{g' \in \mathcal{B}(g, \delta_g)} \mathbb{1}[\pi \text{ achieves success for the goal } g'] \, dg',$$

  where $\mathcal{B}(g, \delta_g)$ indicates a ball with radius $\delta_g$, centered at $g$. And $\delta_g$ is the same threshold using in reward function (1) and success testing.

- The average success rate oracle $f(\pi, g)$ is estimated by $10^2$ samples.

Figure 8: Visualization of modified task distribution in Experiment 5.2. The initial position of the object is as shown in this figure, and the goal is uniformly generated in the blue region.

## C   Implementation Details

### C.1   Hyper-Parameters

Almost all hyper-parameters using DDPG and HER are kept the same as benchmark results, only following terms differ with Plappert et al. (2018):

- number of MPI workers: 1;
- buffer size: $10^4$ trajectories.

Other hyper-parameters:

- Actor and critic networks: 3 layers with 256 units and ReLU activation;
- Adam optimizer with $10^{-3}$ learning rate;
- Polyak-averaging coefficient: 0.95;
- Action $L_2$-norm penalty coefficient: 1.0;
- Batch size: 256;
- Probability of random actions: 0.3;
- Scale of additive Gaussian noise: 0.2;
- Probability of HER experience replay: 0.8;
- Number of batches to replay after collecting one trajectory: 20.

Hyper-parameters in weighted bipartite matching:

- Lipschitz constant $L$: 5.0;
- Distance weight $c$: 3.0;
- Number of hindsight goals $K$: 50 or 100.

### C.2   Details on Data Processing

- In policy training of HGG, we sample minibatches using HER.
- As a normalization step, we use Lipschitz constant $L^* = \frac{L}{(1-\gamma)d^{max}}$ in back-end computation, where $d^{max}$ is the $\ell_2$-diameter of the goal space $\mathcal{G}$, and $L$ corresponds to the amount discussed in ablation study.
- To reduce computational cost of bipartite matching, we approximate the buffer set by a First-In-First-Out queue containing $10^3$ recent trajectories.
- An additional Gaussian noise $\mathcal{N}(0, 0.05I)$ is added to goals generated by HGG in Fetch environments. We don't add this term in Hand environments because the goal space is not $\mathbb{R}^d$.

# D Additional Experiment Results

## D.1 Additional Visualization of Hindsight Goals Generated by HGG

Figure 9: Additional visualization to illustrate the hindsight goals generated by HGG.

To give better intuitive illustrations on our motivation, we provide an additional visualization of goal distribution generated by HGG on a complex manipulation task FetchPickAndPlace (Figures 9a and 9b). In Figure 9a, "blue to green" corresponds to the generated goals during training. HGG will guide the agent to understand the location of the object in the early stage, and move it to its nearby region. Then it will learn to move the object towards the easiest direction, i.e. pushing the object to the location underneath the actual goal, and finally pick it up. For those tasks which are hard to visualize, such as the HandManipultation tasks, we plotted the curves of distances between proposed exploratory goals and actually desired goals (Figure 9c), all experiment followed the similar learning dynamics.

## D.2 Evaluation on Standard Tasks

In this section, we provide experiment results on standard Fetch tasks. The learning are shown in Figure 10.

Figure 10: Learning curves for HGG and HER in standard task distribution created by Andrychowicz et al. (2017).

## D.3 Additional Experiment Results on Section 5.2

We provide the comparison of the performance of HGG and explicit curriculum learning on Fetch-PickAndPlace environment (see Figure 11), showing that the result given in Section 5.2 generalizes to a different environment.

Figure 11: Comparison with explicit curriculum learning in FetchPickAndPlace. The initial position of the object is as shown in the left figure, and the goal is generated in the blue region following the default distribution created by Andrychowicz et al. (2017).

## D.4 Ablation Study

We provide full experiments on ablation study in Figure 12.

Figure 12: A full version of ablation study.