[Reviews · NeurIPS 2019]

Reviewer 1



This work is original, to the best of my knowledge, in automatically grading an agents capability in a Goal directed MDP. This allows the agent to generate increasingly difficult goals to perform by solving a Wasserstein barycenter problem; they effectively explain this problem as solve it well by exploiting some structure in the MDP. The authors also provide a simple, intuitive setting where the effects of HGG are conveyed, but do not include much discussion of their more complex experimental results. While I do not doubt the validity of their approach, when experimental results are included it seems odd to not try and dissect what their method was able to contribute. Their ablation studies (in the main body as well as the supplemental material) offer additional evidence that their approach is indeed valid; one should expect that their automatic method should be robust to a reasonable range of hyper-parameters, which was important to demonstrate.

Reviewer 2



update: I've read the rebuttal and keep my original score. It is good to see new results on a harder environment + how choosing right metric helps in another one. I keep the original score since the theory/math is sufficient for this method, but is not a significant contribution. --- The authors propose a novel curriculum generation technique that uses both the observed task goals/initial states and the visited states information to automatically generate a set of achievable but task-relevant goals for better exploration. The technique is combined with HER to achieve substantial improvements on sparse reward domains, especially in the context with adversarial initial-goal states. The paper is written very cleanly and in good quality. The motivation is well grounded based on the smoothness of value functions. The idea is simple but novel and effective. The illustration in Figure 2 is convincing of the failure of HER in adversarial cases and how this approach can improve exploration. Figure 4’s comparison with prior ground-truth success-based curriculum also shows favorable result. This curriculum method is also realistic, as it does not make resettability assumption or Monte Carlo success rate evaluation as done in other works such as reverse curriculum generation. The author also discusses the core limitation of the method as the reliance on distance metric for curriculum and suggests future work in conclusion. Comments: - extension: for faster adaptation to target goal distribution, authors may even consider changing exploration goal within a rollout (in K=1 setting) - can monotonic improvement guarantee for this curriculum approach be derived?

Reviewer 3



The authors propose a new method for sampling exploration goals when performing goal-conditioned RL with hindsight experience replay. The authors propose a lower bound that depends on some Lipschitz property of the goal-conditioned value function with respect to the distance between the goals and states. The authors demonstrate that across various Fetch-robot tasks, their method, when combined with EBP (a method for relabeling goals), outperforms HER. The authors also perform various ablations that show their method is relatively insensitive to hyperparameter values. Overall, the empirical results are solid, but the math behind the paper is rather troubling. Specifically, the theorem seem rather vacuous: Writing “x” in place of “(s, g)”, the theorem basically says that if V(x1) >= V(x2) + d(x1, x2), then if you take the expectation of both sides (w.r.t. any coupling over x1 and x2), the inequality still holds. Taking the minimum overall couplings gives the theorem. Reading the (brief) proof only makes me more confident that the theorem is not insightful. Math aside, the intuition behind the method is clear (especially with Figure 2). The sensitivity analysis and comparisons seem appropriate, though not particularly amazing. Some smaller comments: The first paragraph of the introduction seems rather generic. It would be good to quickly focus on the problems the authors are trying to solve (goal-directed exploration) more quickly. I do wonder how important it is to solve the bipartite matching problem exactly. For example, could the authors have instead sampled 100 different trajectories and taken the max over all time steps and trajectories? The related works could discuss more work such as [1-7], though some of the work is extremely recent and I do not penalize the authors for not including them. [1] S. Forestier, Y. Mollard, and P.-Y. Oudeyer. Intrinsically Motivated Goal Exploration Processes with Automatic Curriculum Learning. [2] A. Péré, S. Forestier, O. Sigaud, and P.-Y. Oudeyer. Unsupervised Learning of Goal Spaces for Intrinsically Motivated Goal Exploration. [3] C. Colas, et. al. GEP-PG: Decoupling Exploration and Exploitation in Deep Reinforcement Learning Algorithms [4] V. Pong, et. al. Skew-Fit: State-Covering Self-Supervised Reinforcement Learning [5] V. Veeriah. J Oh., and S. Singh. Many-Goals Reinforcement Learning. [6] R. Zhao, et. al. Maximum Entropy-Regularized Multi-Goal Reinforcement Learning [7] Kaelbling, Leslie P. Learning to achieve goals. -- After reading the author feedback, I still find the mathematical results rather weak. Again, the theorem simply says "if the triangle inequality is true for all states, then it is true in expectation, no matter what the (coupling) distribution is. There's no clear explanation for why this assumption would be reasonable, and the only justification provided is that prior related work made this assumption implicitly. Personally, I believe that that it is a reasonable assumption in many applications, and the paper would be strengthened by explicitly discussing this, rather than leaving it to the reader. I'm increasing my score in spite of the distracting mathematical discussion, primarily due to the strong empirical results (performance and visualizations).

[Author Response · NeurIPS 2019]

We thank the reviewers for their constructive comments and helpful feedback.

**More illustrations on difficult tasks (R1,R3)** To give better intuitive illustrations on our motivation, we provide an additional visualization of goal distribution generated by HGG on a complex manipulation task FetchPickAndPlace (left and middle figures). In the left figure, "blue to green" corresponds to the generated goals during training. HGG will guide the agent to understand the location of the object in the early stage, and move it to its nearby region. Then it will learn to move the object towards the easiest direction, i.e. pushing the object to the location underneath the actual goal, and finally pick it up. For those tasks which are hard to visualize, such as the HandManipulation tasks, we plotted the curves of distances between proposed exploratory goals and actually desired goals (right figure), all experiment followed the similar learning dynamics. We will add more intuitive demonstrations in the next version.

**Regarding the difficulty of complex tasks (R1,R2)** HGG does help to improve sample efficiency in difficult tasks, but its *final performance* is still limited by back-end RL algorithm (i.e. DDPG). For example, HGG provides a better warming-up in the experiment of FetchSlide, despite the final accuracy seems to be similar. For complex tasks which DDPG provides reasonable training, HGG also demonstrates large improvement over HER, as shown in the paper. We will add extra analysis and discussion about it.

About the metric we used in this paper, we agree that a crafted or a learned metric may be more suitable than $\ell_2$ for difficult tasks. To show this, we created an environment with an obstacle (left figure). The object and the goal are uniformly generated in the green and the red segments respectively. The brown block is a static wall which cannot be moved. In addition to $\ell_2$, we also construct a distance metric based on the graph distance of a mesh grid on the plane, as shown in the left figure, the blue line is a successful trajectory in such hand-craft distance measure. Intuitively, this distance should be better $\ell_2$ due to the existence of the obstacle. Experimental results (in the right Figure) also suggest that such a crafted distance metric can provide better for goal generation and training, and significantly improve sample efficiency over $\ell_2$ (right figure). It would be a future direction to investigate ways to obtain or learn a good metric.

**Regarding the theorem (R3)** We are sorry about the insufficiency of motivation and our inattention to the presentation formality of this theorem. Our theorem serves as a tool, as well as a justification, to get the objective function that we use, especially the term of Wasserstein distance between distributions of surrogate goals and underlying goals. We also believe that the lower bound we obtained in this theorem, although we agree that it is quite simple to prove, is not absolutely vacuous. For example, when the value function is linear, the lower bound becomes tight and cannot be further improved. We also agree with R3 that there can be also a clear and intuitive motivation of our method. We'll rewrite this part by improving the presentation, make its usage easier to understand, increase the rigorousness of formal mathematical statements, and provide further intuitive explanation of our method in the next version.

**Regarding the Lipschitz condition (R3)** Thank you for the suggestion about the validness of this condition. Similar generalizability assumptions were also required by many previous works on goal curriculum (e.g. two related works mentioned at line 118), they assumed the learned policy could generalize to close-by regions. In our method, Lipschitz continuity is assumed to simplify the introduction of algorithmic framework and objective. It is a formalization of the intuition that, the performance of a policy is similar for tasks that are close to each other. In principle, we require the smoothness of validated policy and environment dynamics to derive an approximated version of Lipschitz continuity. These two preconditions are commonly satisfied in many continuous robotics tasks with parameterized policy classes. We will provide empirical ways to verify this condition on several robotics tasks in the OpenAI gym benchmark.

As suggested by R3, we will include more intuitive discussion of this condition and provide further mathematical and practical guidelines of it in the next version.

**Response to other helpful suggestions (R3)** Thank you for pointing out the related references and suggesting comparison to simpler optimization schemes. We will add the suggested related works and discussions in the next version. We will investigate a simpler implementation which chooses the argmax goal instance in a random subset of trajectories, as suggested. Some preliminary results indicate that it achieves reasonable performance on simple manipulation tasks. We will add comprehensive results on this part in the next version.

[Meta-Review · NeurIPS 2019]

Following the authors' rebuttal, there is a consensus to accept the paper. The authors are requested to address the reviewers' concerns and integrate material from the rebuttal into the revision.